# Exploring the Impact of Corporate Social Responsibility on Financial Performance: The Moderating Role of Media Attention

Jiangjun Li, Tao Fu *, Shengyue Han and Rui Liang

School of Urban Economics and Management, Beijing University of Civil Engineering and Architecture, Beijing 102600, China
* Correspondence: futaowssy@163.com

**Abstract:** In the post-epidemic era, more and more enterprises have realized the crucial significance of corporate social responsibility for enterprise development. However, there is no consensus on the relationship between CSR and financial performance (FP). We collected data on listed companies in China from 2014 to 2020 in order to demonstrate whether CSR is positively or negatively correlated with financial performance and studied this relationship for the first time using media attention as a moderating variable. Through a regression analysis, we found that (1) companies with good CSR performance show a high level of FP; (2) the higher the media's attention on the company, the better the CSR performance; and (3) based on the nature of the emotion, we divided media attention into positive and negative reports. Positive reports weaken the positive impact of CSR on financial performance, while negative reports reinforce this positive effect. These empirical findings remain robust after controlling for endogeneity and employing alternative variable measures. The results in this paper complement recent nexus modeling work and give a better understanding of the interaction mechanism in the CSR-FP nexus with useful implications for future enterprises' sustainable development.

**Keywords:** corporate social responsibility (CSR); performance improvement; moderating effect; media attention; listed firms in China

## 1. Introduction

Nowadays, fierce market competition poses a challenge to corporate sustainability [1]. In order to achieve sustainable corporate development, companies should not only be responsible for profits but also for the environment and assume the corresponding social responsibilities [2]. Since the early 2000s, CSR has attracted widespread attention from academia and society, and has become a hot topic in the media and among investment communities, regulators, and the public [3]. International organizations such as the Council on Economic Compact (CEPAA), United Nations Global Compact (UNGC), and International Organization for Standardization (ISO) have published CSR standards. This marks the fact that CSR has officially entered the international perspective. Some developed countries have issued relevant laws and regulations to promote the implementation of CSR. The focus of CSR is voluntary, which goes beyond the legal and contractual obligations of the company [4]. It requires enterprises to pay more attention to the needs of other stakeholders while pursuing interests and creating value. In fact, since the COVID-19 outbreak, active participation in social responsibility activities has become the primary task of modern enterprises [5]. Activities related to CSR include protecting employees' health, developing science and technology, participating in welfare investment, charity, and environmental protection. With the deepening of economic development and social division of labor, more and more research has concluded that CSR meets the needs of stakeholders, enhances

and improves corporate reputation and financial status, and adds weight to the long-term survival of enterprises in the market [6].

China is the largest developing country in the world [7]. Chinese companies are contributing more and more to the international market. Although China has turned to a stage of high-quality development, a series of problems, such as medical health, food safety, and elderly care have become increasingly prominent, and enterprises have a weak awareness of fulfilling and disclosing CSR. The "Guidelines on Social Responsibility of Listed Companies", "Guidelines on the Application of Internal control of Enterprises", "China Corporate Law", and "Environmental Information Disclosure of Listed Companies Guide" issued by relevant departments are all guiding enterprises to actively participate in CSR activities [8]. Although the development of CSR in China has started late, as time goes by, more and more enterprises have begun to take the initiative to undertake and disclose corporate social responsibility. Especially during the outbreak of COVID-19 in 2020, many enterprises (such as Alibaba, Bosidegn, Kingclean, Tofflon, etc.) rushed to Wuhan to donate money and goods. According to the social responsibility rating issued by RKS, an authoritative social responsibility rating agency in China, in 2018, 2019, 2020, and 2021, there were 888, 987, 1112, and 1366 listed companies that issued social responsibility reports, respectively. The public is paying more and more attention to enterprises, which makes enterprises pay more attention to social responsibility issues. With the development of digital technology, the public can access corporate information more easily through social media. This means that the media's attention to CSR activities will affect the company's business decision-making and thus the company's financial performance.

Although corporate social responsibility has received extensive attention, can the implementation of corporate social responsibility bring benefits to enterprises? The relationship between CSR and financial performance has been vigorously discussed in academia; however, there are three different views: First, excessive investment in CSR activities will reduce the opportunities to maximize profits by using resources [9]. In addition, engaging in CSR activities will increase costs because it will lead to conflicts among stakeholders, thus reducing corporate performance [10,11]. Second, undertaking more CSR can improve corporate financial performance [12–15]; therefore, many large companies worldwide invest significant resources to actively undertake social responsibility, and they disclose their CSR activities to stakeholders and potential investors through various channels such as annual sustainability reports [16]. Third, there is a nonlinear relationship between CSR and FP [17]. There has been extensive research on the relationship between the two, but due to the different data and methods of the research, the research conclusions are not the same [18–20].

This paper re-examines the relationship between CSR and FP in the context of China. In this process, we further explore the boundary conditions of the relationship between CSR and FP based on existing studies. Enterprises may be affected by external factors when carrying out social responsibility activities. Therefore, this paper studies how positive and negative reports affect the relationship between CSR and FP differently. Specifically, we use Rankins CSR Ratings (RKS) to measure CSR performance. RKS independently developed the first CSR report rating system in China, which is the authoritative third-party rating agency of CSR in China and is committed to providing scientific and objective corporate responsibility rating information for investors, consumers, and the public. When the FP indicators are replaced in the robustness test, the results are still reliable.

This paper differs from previous studies in several aspects. Firstly, the object of study. Under the development pattern of "domestic and international circulation", China's economic development pays more attention to green, environmental protection and humanistic care, and Chinese people also pay more attention to corporate social responsibility. Therefore, taking China as the research object is of great significance for the study of CSR and FP. Secondly, the moderating role. Although some scholars have studied the influence of media attention on CSR [21] and the influence of CSR on FP [22], they give little attention to the moderating role of media attention in the relationship between CSR and FP. This is

the original value of this study: that media attention and the impact of CSR on FP have been examined jointly within the context of China, offering useful implications for future enterprises' sustainable development. Lastly, financial performance indicators. Different from Achim's research, this paper reflects the financial performance of enterprises by using profit margin before interest and tax on total assets excluding earnings management, which can not only make the financial performance more real and reliable, but also better reflect the operational capacity, asset utilization efficiency, and financial management level of enterprises. The following contents of this paper are organized as follows: the second part is the literature review and research hypotheses, the third part is the research methods, and the fourth part is the analysis of the empirical results and the robustness test. Finally, some discussions are presented as the conclusion of this paper.

## 2. Literature Review

Through the literature review, we found that scholars have been carrying out extensive research on CSR and FP.

### 2.1. CSR-FP

The first to formally propose CSR was Sheldon. In 1924, he pointed out that when an enterprise pursues the maximization of interests, it should also pay attention to the needs of relevant groups inside and outside the enterprise and contribute to society. However, after nearly a century of theoretical and practical development, there is still no clear and accurate definition of CSR. From the perspective of economics, it is believed that the greatest responsibility of an enterprise is to continuously pursue profits within the scope stipulated by law [23]. From the perspective of sociology, while being responsible to shareholders and earning profits, enterprises should also participate in social welfare protection and pay attention to people's livelihood and ecological construction. Howard Bowen, known as the "father of corporate social responsibility", also pointed out that the social responsibility of managers is to enact policies, make decisions, and take actions according to social standards and values. In this vein, we can think that CSR is not only the obligation of enterprises to pursue long-term goals but also an important part of socially sustainable development [24]. CSR is also important for enterprises to disclose non-financial information, to effectively alleviate information asymmetry, strengthen internal control, and avoid insider trading. The growing importance of CSR to corporate and social development makes it an important research topic in management and accounting [25,26]. Managers use resources prudently, and only those activities that contribute to CSR can help companies gain competitive advantages [27]. In their study, Hu et al. pointed out that corporate behavior that enhances stakeholder value has become a new measure of financial performance [28]. Looking at the relevant literature, it can be found that since the 1970s, the research on the relationship between CSR and FP has been a very hot topic [29].

However, the results of their studies are quite different. There are several statements such as positive correlation, negative correlation, irrelevance, and nonlinearity. Studies using different CSR measures and financial performance variables will yield different results [30]. In existing studies, most scholars believe that CSR can promote the improvement of financial performance [31–34]. Chen and Wang found that, in the Chinese market, fulfilling corporate social responsibility can improve the financial performance of the current year and the next year [35]. Maqbool and Zameer demonstrated that CSR can improve the profitability and stock returns of Indian banks. Cochran and Wood found that if a company can actively implement CSR in the normal production and operation processes, the financial performance of the company will be improved compared with the company that does not implement CSR [36]. Stakeholder theory puts forward the goals and content of corporate social responsibility and posits that, through corporate social responsibility activities, the conflicts of interest between stakeholders and enterprises can be reduced [37], thereby establishing a good reputation for enterprises, enabling enterprises to reduce costs or gain differentiated competitive advantages to improve their financial performance [38,39]. How-

ever, some scholars believe that it is unwise to invest in a field unrelated to the operation of the enterprise, which is a waste of enterprise resources [7]. Agency theory argues that CSR activities are the manifestation of excessive investment by managers. Managers will lose corporate profits to improve their own reputation or that of enterprises, thus weakening corporate performance [40,41]. Therefore, enterprises should pay more attention to how to improve their operational efficiency, instead of wasting resources on social organizations that have nothing to do with their operations. In addition, some scholars believe that there is no relationship between CSR and FP [42], or that there is a nonlinear relationship between the two [43,44].

In China, incidents such as "Poisonous Powdered Milk ", "Clenbuterol", and "Foxconn's Employee Jumping off the Buildings" have aroused the public's in-depth thinking on corporate social responsibility. Once an enterprise performs its social responsibility improperly, it will damage its reputation at least and affect its performance at worst, possibly even leading to its bankruptcy. Therefore, in the past decade, many enterprises have paid more and more attention to the important role of corporate social responsibility, which can promote the sustainable development of enterprises. Based on this, we constructed hypothesis H1:

**Hypothesis 1 (H1).** *Higher CSR activities are associated with higher FP.*

### 2.2. Media Attention–CSR

There are various reasons for enterprises to carry out social responsibility activities [45]. In the process of enterprise management, one of the problems that managers must solve is to reduce stakeholders' suspicion of CSR activities. In other words, the difficulty that managers need to overcome is to communicate with stakeholders smoothly [46]. Chae pointed out that, although some companies will disclose their social responsibility activities, consumers still tend to learn about products or brands through third-party platforms or media reports when purchasing products [47]. Therefore, enterprises need to use media to spread corporate information. As a tool for disseminating and amplifying information, we should not underestimate the power of the media in the information age. In a complex market environment, the media can disseminate information about corporate social responsibility quickly and widely, and corporate stakeholders can maximize the use of this information to make investment decisions and improve market competitiveness [48]. It can be seen that media attention, with its timeliness and authenticity, plays a key role in corporate social responsibility and corporate performance, motivating enterprises to pay attention to social responsibility to improve their own performance.

In addition, media attention has played an external regulatory role for listed companies, and it is also an important way for ordinary citizens to obtain information about companies' fulfillment of social responsibilities [49]. As an independent third party, the media actively reports the news of corporate fulfillment of social responsibility, which improves the visualization and transparency of corporate behavior. Therefore, both positive and negative reports force listed companies to pay more attention to the implementation of CSR [50]. The media amplifies the influence of CSR behavior. When the media reports positively on corporate social responsibility activities, it can more effectively portray a good image of the company, thereby attracting investors to invest; when the media reports negatively on social responsibility activities, the company will be under the pressure of public opinion. At this time, it will increase investment in social activities to build good reputation capital. By combing through previous studies, Dyck found that the supervision of media attention on enterprises can reach 13% [51]. It can be seen that media attention can indeed influence corporate behavior. Because media attention reduces the information asymmetry between consumers and other stakeholders and enterprises, listed companies are more willing to take the initiative to disclose CSR to safeguard their own honor and corporate interests.

Therefore, if the media can report objectively and fairly, CSR activities can win more trust from consumers [52]. Companies that actively undertake social responsibilities will actively seek media attention and expect positive reports. When the media carries out positive reports on the social activities of enterprises, the public will grasp more information about enterprises, enhancing enterprises' willingness to disclose information, thus forming a positive cycle [53].

Based on the above-mentioned viewpoints in the literature, we believe that the media can play a role in the supervision of external public opinion on enterprises under the condition of respecting objective facts, which is beneficial to corporate governance and the effective use of resources. Therefore, we proposed the second hypothesis:

**Hypothesis 2 (H2).** *There is a positive relationship between media attention and CSR activities.*

*2.3. The Moderating Role of Media Attention*

In recent years, there has been abundant research on CSR and FP as well as media attention and CSR, but only some research combines the three. With the continuous development of information technology, the media of information transmission is increasingly diversified, and the utilization rate of media is constantly improving. Media, as a third party, will report corporate information without bias, making it gradually become an important carrier for stakeholders and enterprises to communicate information [54]. Scholars such as Abbas found that the performance level of heavily polluting enterprises (power industry) sensitive to media reports will be greatly improved when they use low-pollution equipment [55]. However, these scholars all focus on media attention as a mediating variable. In fact, in the research on the relationship between CSR and FP, there is no consistent conclusion about the strength and direction of influence between the two. The more media attention focused on a company, the greater the external pressure on the company [56]. Therefore, media attention can greatly promote the autonomy of enterprises, making them attach great importance to the social impact of their operations. Stakeholders understand and monitor the actions of enterprises through media reports and "reward and punish enterprises" through "purchase or not"; the results are directly reflected in the financial performance of enterprises [57]. In addition, the media, as a tool to guide trends in public opinion in society, will exert binding force and external pressure on the behavior of enterprises, prompting enterprises to be responsible for the social impact of their various business behaviors, thus affecting enterprise performance. Therefore, from a statistical point of view, it seems more convincing to study media attention as a moderating variable. In this vein, we proposed the third hypothesis:

**Hypothesis 3 (H3).** *The relationship between CSR and FP is moderated by media attention.*

The conceptual framework for the study is shown in Figure 1.

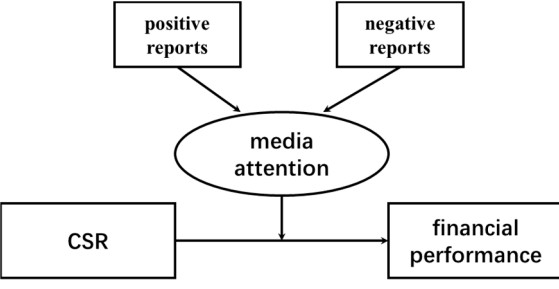

**Figure 1.** Proposed research model.

## 3. Methodology

### 3.1. Sample and Data Sources

Under the joint efforts of multiple forces such as the government, capital markets, and industry associations, the disclosure of CSR in China had an "explosive" growth in 2014. Listed companies accounting for 75.3% of the total disclosures, becoming the main force behind the publication of CSR reports in China. Therefore, we selected Chinese listed companies from 2014 to 2020 as the research sample. The sample was composed of unbalanced panel data, including 432 valid samples and 2964 observations after excluding financial and insurance companies, ST and *ST companies, as well as companies with extreme, abnormal, and missing data. The CSR score in this article comes from the rating report of RKS, the third-party evaluation agency. RKS is based on a hundred-mark system: the higher the score is, the more effective the company's CSR work is. Other data such as financial performance, debt leverage, cash flow, Tobin's q, ownership concentration, nature of ownership, and education development are sourced from China Stock Market and Accounting Research Database (CSMAR). GDP comes from the National Bureau of Statistics of China. To avoid the influence of extreme values on the research conclusions, all variables are winsorized at the 1% level. The relevant data were processed manually by EXCEL and statistically by STATA.

### 3.2. Measures

#### 3.2.1. Independent Variable

We take corporate social responsibility (CSR) as the core explanatory variable. Given the reliability and availability of data, we adopt the rating reports issued by RKS, which are authoritative and widely recognized by academic and social circles. Through structured expert scoring, the rating reports of RKS can comprehensively reflect the performance of CSR.

#### 3.2.2. Dependent Variable

We take financial performance (FP) as the explained variable. Considering the accuracy of accounting information, financial performance is generally directly expressed by the profit margin before interest and tax on total assets or the return on equity, and less consideration is given to the negative effect of earnings management. Therefore, we learn from Zhao Lijuan's practice [58]: Unebit (profit margin before interest and tax on total assets excluding earnings management) is used to express financial performance.

#### 3.2.3. Moderator Variable

We take media attention as a moderating variable and measure it by the total number of online media news reports. The data processing methods are as follows: Firstly, the company's abbreviation is entered into the advanced search of Baidu's news search engine. Then, before the date of each annual audit report, the report containing the company's abbreviation in the title is searched. After that, we filter out announcements from listed companies, analyst recommendations, and reports involving multiple listed companies in a single report. Finally, the total number of relevant news reports is counted as the amount of media attention. In terms of negative reports, whether there are obvious negative evaluation words or tone in the title and content of the report is used as the basis for negative reports, such as "scandal", "loss", "violation", "warning", and other words. While neutral reporting is ostensibly "neither good nor bad," it increases corporate visibility in society. Therefore, we treat neutral reporting as positive.

#### 3.2.4. Control Variables

Based on related research on corporate social responsibility, media attention, and financial performance, we selected the company's debt level, cash flow, Tobin's q, ownership concentration, ownership nature, educational development, and regional economic development level as control variables.

In the process of data collection, considering the availability and reliability of the data, the asset–liability ratio is used to replace the debt level. Net cash flow (operating activities/total assets) is used to replace cash flow. The shareholding ratio of the largest shareholder is used to replace ownership concentration. The number of "211" and "985" universities in the province where the company is registered is used to replace the level of educational development. Regional GDP is used to replace the level of regional economic development. The nature of ownership is represented by Nature, with 1 meaning state-owned enterprises and 0 meaning non-state-owned. We summarize the variable definitions, including attributes, descriptions, and measurements, in Table 1.

**Table 1.** Description of variables used in this study.

| Variable | Symbol | Definition |
|---|---|---|
| Core variables | | |
| Financial Performance | Unebit | Total assets EBITDA margin after excluding surplus management |
| Corporate Social Responsibility | CSR | 2014–2020 CSR ratings (based on the rksratings.cn database) |
| Media Attention | Me $\quad$ Me$_1$ $\quad$ Me$_2$ | Total media coverage $\quad$ Positive media coverage $\quad$ Negative media coverage |
| Control variables | | |
| Debt Leverage | Lev | Corporate assets-to-liability ratio |
| Cash Flow | Cash | Net cash flows from operating activities/total assets |
| Tobin's Q Ratio | Q | 2014–2020 (based on the ccerdata.cn database) |
| Ownership Concentration | Top | Number of shares held by the largest shareholder divided by the total share capital of the company |
| Nature of Ownership | Nature | A dummy variable that takes value 1 if state-owned, and 0 otherwise |
| Education Development | Education | Number of top universities in the province where the company is registered |
| Gross Domestic Product | GDP | 2014–2020 GDP (based on the stats.gov.cn database) |

### 3.3. Model Building

By extensively reading the research on the relationship between CSR and FP, we use the econometric model of Xiong [45] and cluster standard errors by firm to verify the proposed hypotheses.

To test H1 proposed above, model (1) is constructed as follows.

$$\text{Unebit}_{i,t} = \alpha_0 + \alpha_1 \text{CSR}_{i,t} + \alpha_2 \text{Lev}_{i,t} + \alpha_3 \text{Cash}_{i,t} + \alpha_4 Q_{i,t} + \alpha_5 \text{Top}_{i,t} + \alpha_6 \text{Nature}_{i,t} + \alpha_7 \text{Education}_{i,t} + \alpha_8 \text{GDP}_{i,t} + \varepsilon_1 \quad (1)$$

To test H2 proposed above, model (2) is constructed as follows.

$$\text{CSR}_{i,t} = \alpha_0 + \alpha_1 \text{Me}_{i,t} + \alpha_2 \text{Lev}_{i,t} + \alpha_3 \text{Cash}_{i,t} + \alpha_4 Q_{i,t} + \alpha_5 \text{Top}_{i,t} + \alpha_6 \text{Nature}_{i,t} + \alpha_7 \text{Education}_{i,t} + \alpha_8 \text{GDP}_{i,t} + \varepsilon_2 \quad (2)$$

To test H3 proposed above, model (3) is constructed as follows.

$$\text{Unebit}_{i,t} = \alpha_0 + \alpha_1 \text{CSR}_{i,t-1} + \alpha_2 \text{Me}_{i,t-1} + \alpha_3 \text{Csr}_{i,t} \times \text{Me}_{i,t-1} + \alpha_4 \text{Lev}_{i,t-1} + \alpha_5 \text{Cash}_{i,t-1} + \alpha_6 Q_{i,t-1} + \alpha_7 \text{Top}_{i,t-1} + \alpha_8 \text{Nature}_{i,t-1} + \alpha_9 \text{Education}_{i,t-1} + \alpha_{10} \text{GDP}_{i,t-1} + \varepsilon_3 \quad (3)$$

In Equations (1)–(3), $\alpha$ is the estimated coefficient. In addition, i represents enterprise; t represents time; $\varepsilon$ is the random error term. In order to avoid endogeneity, this paper will be explained by a one-year lag; that is, all variables are the values of the previous year corresponding to FP.

## 4. Discussion

### 4.1. Statistics Description

Table 2 shows the descriptive statistics of the sample. It can be seen that the mean value of CSR is 41.932 and the standard deviation is 11.536. The large gap between the maximum value of 78.529 and the minimum value of 21.649 indicates that different enterprises have uneven CSR performance. Some enterprises are more active, while some enterprises need to further improve their CSR performance. There is a large difference in Unebit, with a minimum value of $-1.145$ and a maximum value of 0.469, indicating a gap in sample enterprises' performance. The average level of Lev is 49.55%, and the overall level of financial leverage is relatively appropriate, but there is a large difference between the minimum value of 7.49% and the maximum value of 86.49%, indicating that the financial leverage of the sample companies is very different. The standard deviations of Cash and Q are 0.059 and 1.224, respectively. From the perspective of the shareholding ratio of the largest shareholder, the average value is 38.27, indicating that the sample enterprises have a high shareholding concentration. There is a large difference between the maximum value of 32 and the minimum value of 1, which indicates that the level of education development in different provinces is not balanced. There is little difference between the median and mean of GDP, and the overall distribution is more even.

**Table 2.** Statistical description of all variables.

| Variable | Obs | Mean | Std. Dev. | Min | Max | P50 |
|----------|-----|------|-----------|-----|-----|-----|
| CSR | 3024 | 41.932 | 11.536 | 21.649 | 78.529 | 39.232 |
| Unebit | 3024 | 0.04 | 0.087 | $-1.145$ | 0.469 | 0.038 |
| Lev | 3024 | 49.55 | 19.027 | 7.493 | 86.488 | 51.019 |
| Cash | 3024 | 0.057 | 0.059 | $-0.118$ | 0.228 | 0.055 |
| Q | 3024 | 1.4088 | 1.224 | 0.134 | 7.133 | 1.024 |
| Top | 3024 | 38.27 | 15.301 | 8.14 | 77.32 | 38.07 |
| Education | 3024 | 8.762 | 10.212 | 1 | 32 | 5 |
| GDP | 3024 | 10.324 | 0.625 | 7.79 | 11.485 | 10.264 |

### 4.2. Correlation Analysis

It can be seen from Table 3 that the correlation coefficients between Unebit and CSR, Lev, Cash, Q, Top, Nature, and GDP are all significant. Specifically, Unebit has a significant positive correlation with CSR, Cash, Q, Nature, and GDP at the 1% level. Unebit showed a significant negative correlation with Lev at the 1% level, and there was no significant correlation with education. To avoid distortion of the significance of the empirical results, we compared the correlation coefficients among variables.

To ensure that there is no multicollinearity among the study variables, before the regression analysis, we refer to the research method of Anderson [59] to test the multi-collinearity problem: if the correlation coefficient between the two variables is greater than 0.65, it means that there is a collinearity problem between the two variables. As shown in Table 3, the maximum correlation coefficient between any two variables is 0.4342, so it is considered that it cannot form collinearity.

**Table 3.** Correlations analysis.

| Variable | CSR | Unebit | Lev | Cash | Q | Top | Nature | GDP |
|---|---|---|---|---|---|---|---|---|
| CSR | 1.0000 | | | | | | | |
| Unebit | 0.0885 *** | 1.0000 | | | | | | |
| Lev | 0.1268 *** | −0.1584 *** | 1.0000 | | | | | |
| Cash | 0.1069 *** | 0.4342 *** | −0.2587 *** | 1.0000 | | | | |
| Q | −0.1128 *** | 0.1669 *** | −0.5993 *** | 0.2190 *** | 1.0000 | | | |
| Top | 0.1063 *** | 0.0512 ** | 0.0360 | 0.0911 *** | −0.0880 *** | 1.0000 | | |
| Nature | 0.1278 *** | −0.0769 *** | 0.1803 *** | −0.0586 ** | −0.2045 *** | 0.3283 *** | 1.0000 | |
| GDP | 0.0826 *** | 0.1200 *** | −0.0408 | 0.0682 *** | 0.0264 | −0.0978 *** | −0.2218 *** | 1.0000 |

** $p < 0.05$; *** $p < 0.01$. Two-tailed.

*4.3. Regression Results*

4.3.1. Regression Analysis of CSR and FP

Table 4 shows the regression analysis results of model variables. The result shows that there is a significant positive correlation between CSR and Unebit at the 10% level (r = 5.405, $p < 0.1$), which is consistent with H1. This means that the enterprise cannot exist independently in society because they are always inextricably linked with society and other stakeholders. Only by earnestly fulfilling its social responsibilities can it establish a good corporate image and attract more investors to cooperate to make the enterprise obtain good profits. At the same time, the corrected coefficient $R^2$ was 0.072, indicating a good fit between the explaining variable (CSR) and the explained variable (Unebit).

**Table 4.** CSR–Unebit relationship using OLS analysis.

| Independent Variable | Unebit (t) |
|---|---|
| constant | 11.401 *** (2.98) |
| CSR | 5.405 * (1.93) |
| Lev | 0.062 *** (4.59) |
| Cash | 21.921 *** (5.45) |
| Q | −0.615 *** (−3.20) |
| Top | 0.019 (1.21) |
| Nature | 3.161 *** (4.46) |
| Education | 0.069 *** (2.93) |
| GDP | 2.265 *** (6.48) |
| Adj-$R^2$ | 0.072 |

* $p < 0.1$; *** $p < 0.01$.

4.3.2. Regression Analysis of Media Attention and CSR

As shown in Table 5, the correlation coefficient between Me and CSR is 0.259 and there is a significant positive correlation at the 1% level (r = 0.259, $p < 0.01$), which also supports H2 as expected. The corrected coefficient $R^2$ is 0.092, indicating that the model has passed the significance test and has a good fitting effect. This shows that media attention plays an important role in the process of enterprises fulfilling social responsibility. The more the media reports on the company, the more open and transparent the company's information will be and the better the stakeholders' understanding of the company's

operating conditions and development prospects will be. Therefore, more consumers will be attracted to purchase enterprise products, and suppliers and investors will be encouraged to cooperate. When enterprises receive benefits, they will be more active in undertaking and disclosing corporate social responsibility. On the contrary, if the company's exposure is lower, there will be a lack of communication and understanding between society and the company.

**Table 5.** Media–CSR relationship using OLS analysis.

| Independent Variable | CSR (t) |
|---|---|
| constant | 20.337 *** (5.72) |
| Me | 0.259 *** (6.61) |
| Lev | 0.057 *** (4.16) |
| Cash | 24.685 *** (7.41) |
| Q | −0.591 *** (−3.15) |
| Top | 0.089 *** (6.19) |
| Nature | 1.517 *** (3.00) |
| Education | 0.120 *** (5.50) |
| GDP | 1.179 *** (3.66) |
| Adj-$R^2$ | 0.092 |

*** $p < 0.01$.

### 4.3.3. The Impact of Media Attention on This Relationship

To verify that media attention plays a moderating role in the relationship between CSR and FP, we introduce the cross variable $CSR_{i,t} \times Me_{i,t}$ based on the above research, and the regression results are shown in Table 6. As mentioned above, it is known that Me, as a moderating variable, can affect the direction and intensity of CSR towards Unebit. In Table 6, the coefficient of the interaction term CSR*Me is −0.961, which is significant at the 10% level, indicating that media reports play a negative moderating role in the promotion effect of CSR on FP. Additionally, we have divided media attention into positive and negative reports. In Table 6, the regression coefficients of the interaction terms of CSR*$Me_1$ and CSR*$Me_2$ are −1.386 and 13.706, respectively. It can be seen that positive reports play a negative moderating role in this relationship, while negative reports play a positive moderating role.

### 4.4. Robustness Test

To ensure the reliability of the results and address the estimation bias that may be caused by measurement errors, robustness analysis is carried out in this section. When measuring enterprise FP, net profit (np) is used as a substitute variable for FP to analyze the robustness of the model. Net profit, also known as after-tax profit or net income, is the final result of an enterprise's operation and the main index to measure the operating efficiency of an enterprise. Table 7 shows the results of the robustness analysis, which are consistent with the regression model results in Table 4; that is, CSR is positively correlated with financial performance. Therefore, the estimation results of the research model in this paper have good robustness, and the research conclusions have certain universality.

**Table 6.** Moderation result.

| Variable | CSR | | |
|---|---|---|---|
| Unebit | 8.617 *** <br> (2.44) | 9.492 *** <br> (2.70) | 4.024 <br> (1.26) |
| CSR $\times$ Me | −0.961 * <br> (−1.82) | | |
| CSR $\times$ Me$_1$ | | −1.386 ** <br> (−2.31) | |
| CSR $\times$ Me$_2$ | | | 13.706 *** <br> (2.63) |
| N | 2698 | 2698 | 2698 |
| F | 20.74 *** | 21.41 *** | 21.31 *** |
| Adj-R$^2$ | 0.077 | 0.080 | 0.081 |

* $p < 0.1$; ** $p < 0.05$; *** $p < 0.01$.

**Table 7.** Substitute variables for Unebit.

| Independent Variable | np <br> (t) |
|---|---|
| constant | 15.307 *** <br> (4.06) |
| CSR | 0.065 *** <br> (9.96) |
| Lev | 1.986 *** <br> (4.13) |
| Cash | 16.797 *** <br> (4.63) |
| Q | −0.279 <br> (−1.47) |
| Top | −0.011 <br> (−0.75) |
| Nature | 3.174 *** <br> (6.70) |
| Education | 0.017 <br> (0.77) |
| Adj-R$^2$ | 0.141 |

*** $p < 0.01$.

## 5. Discussions and Implications

We selected the data of A-share listed companies from 2014 to 2020 as research samples, collected and sorted the data published by RKS, CSMAR, and the National Bureau of Statistics of China, and then conducted data analysis. In addition, media attention is introduced as a moderating variable to explore its influencing mechanism.

The relationship between CSR and FP has been studied in many organizations and countries. This study complements the CSR literature in developing countries. Enterprises carry out various business activities such as production, sales, and services in society, and always have some kind of connection with all parties in society. If the wishes of all parties cannot be met, the development of enterprises will, in turn, be hindered. Socially responsible businesses tend to be better able to meet the needs of users and consumers in their products and services. For example, providing better employee welfare and adopting green design can improve the consumption utility of users of products and services, thus enabling enterprises to gain advantages over competitors. At the same time, if an enterprise wants to gain long-term competitiveness, it should persevere to promote its social responsibility goals, so as to establish a good long-term image of the enterprise and gain goodwill. Goodwill will eventually lead to good FP, leading to growth in corporate performance. Therefore, putting resources into CSR activities is considered an investment rather than an expenditure. It also precisely verifies that when enterprises actively perform

social responsibility activities, they tend to show a higher level of performance (H1). Similar to our study, Thuy and Khuong et al. [60] proposed that companies that fulfill social responsibilities can build trusting relationships with stakeholders, thereby reducing costs or increasing revenue. Magdalena and Malgorzata [61] proposed that corporate social responsibility is one of the main factors that promotes the improvement of corporate reputation and, then, positively affects financial performance. According to Carvalho and Madaleno [62], social responsibility is the most basic practice of an enterprise. It can not only benefit the public and make the environment sustainable but can also significantly improve an enterprise's asset rotation rate and financial autonomy.

Media, as an important alternative mechanism of the legal system, plays an important role in CSR disclosure. Firstly, it has a supervisory function: the "magnifying glass" function of the media can conduct external supervision on the normal operation of the company, effectively preventing companies from committing financial fraud, regulating their management behavior, and promoting their legal operation. Secondly, it has a function of information transmission: the media is neither the owner nor the stakeholder of the company but an independent third party, so the disclosure of the media has unique independence and legitimacy. The media will process the information obtained from the interview, investigation, and collection and then pass it to the public, which can effectively alleviate the problem of information asymmetry. When the media report more on the enterprise, the stakeholders will know more about the enterprise. In order to maintain its reputation and image, the enterprise will make efforts toward employee welfare, environmental protection, charity, and other aspects [63]. On the contrary, the less media coverage there is of a business, the more unfamiliar it becomes to the public. In order to avoid mistakes in decision-making, investors will adopt a more cautious investment attitude, and enterprises are resistant to stakeholders, which makes conservative enterprises more reluctant to undertake social responsibilities. Therefore, there is a significant positive correlation between media attention and CSR (H2). As Shen [64] said, in the new digital era, thanks to the use of the Internet and social media, the social activities of enterprises are becoming more and more open. When the media carries out in-depth reports on enterprises, the public will reasonably estimate and judge the social activities of enterprises [65]. When enterprises receive more media attention, the promotion effect on enterprises to fulfill social responsibility is more obvious [45].

In the empirical analysis, we conclude that media attention plays a moderating role between CSR and financial performance (H3). To figure out how media attention is moderated, we divided media attention into positive and negative reports. In Table 6, the coefficient of the interaction term CSR $\times$ Me$_1$ is $-1.386$, which indicates that positive reports weaken the positive promotion effect of CSR on financial performance. It is not difficult to understand that enterprises with good performance in social responsibility and strong financial performance have scientific and efficient management systems, and their systems and policies are near perfect. The enterprises will voluntarily participate in social services, and their financial performance will be improved accordingly [13]. For a company in the start-up and growth stage, its biggest goal is how to make the company survive. This kind of CSR performance has been poor in the past. Therefore, when enterprises spend funds to vigorously promote themselves, perform well in public relations, and ensure corporate responsibility to establish a good image for enterprises, it will make small enterprises with insufficient funds struggle, thus reducing their performance. Then, from Table 6, we can see that the interaction coefficient of CSR*Me$_2$ is 13.706, which means that negative reports can play a positive role in promoting the relationship between CSR and financial performance. As mentioned above, the media, in exercising its functions, can supervise enterprises. When there is a problem with the operation, finance, personnel, etc. of the enterprise, the media will question or criticize it. At this time, to stabilize public opinion, the enterprise will make corresponding adjustments and optimizations within the enterprise to promote the enterprise to move forward steadily.

This study has the following important implications. Firstly, this paper enriches the existing literature on the relationship between CSR and FP. In the face of social tragedy, it is of great significance for enterprises to assume social responsibility for their sustainable development. For example, ERKE, a Chinese sportswear company, donated huge sums of money during the floods in Henan province, even as it struggled to survive. This gained the public's respect for ERKE, so they began to engage in "wild consumption", bringing ERKE back from the brink. Second, this is the first study to explore the relationship between CSR and FP using media attention as a moderator variable. The theory and practice of corporate social responsibility in China are still in the early stage of development, and there is still much room for development. The media can not only motivate enterprises to carry out social activities, but also play an indispensable role in the process of stakeholders' understanding of enterprises. Therefore, under the attention of the media, managers and the government can formulate more targeted policies and suggestions and promote enterprises to actively fulfill their social responsibilities to promote the vigorous development of enterprises in the Chinese market.

However, there is no doubt that this study also has the following shortcomings: First of all, we only focus on the listed companies that have disclosed CSR in China, without considering the unlisted or small- and medium-sized companies, and have not included other countries in the scope of this study. Second, when examining the relationship between CSR and financial performance, only the moderating effect of media attention is considered. The relationship between the two is affected by many factors, which also provides opportunities for future research. Therefore, to address the limitations of the study, future studies could collect data on unlisted and small- and medium-size businesses and expand the sample of analysis to additional countries.

**Author Contributions:** Conceptualization, J.L.; methodology, T.F.; software, S.H.; validation, R.L.; data curation, S.H.; writing—original draft preparation, T.F.; writing—review & editing, J.L.; supervision, R.L.; project administration, T.F.; funding acquisition, J.L. All authors have read and agreed to the published version of the manuscript.

**Funding:** This research was funded by CITIC Reform and Development Research Foundation(H21319), the General Projects of Science and Technology Plan of Beijing Municipal Commission of Education (KM202110016006) and Beijing Advanced Innovation Center for Future Urban Design, Beijing University of Civil Engineering and Architecture (UDC2019021424).

**Institutional Review Board Statement:** Not applicable.

**Informed Consent Statement:** Not applicable.

**Data Availability Statement:** The data that support the findings of this study are available from the corresponding authors upon reasonable request.

**Conflicts of Interest:** The authors declare no conflict of interest.

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
