# Peer review of "Exploring the Impact of Corporate Social Responsibility on Financial Performance: The Moderating Role of Media Attention"

_sustainability, doi:10.3390/su15065023_

Round 1

Reviewer 1 Report

1-     The research question is not clear. Make strong discussions about why the research is important, what motivations are required to complete the research and what gaps you want to search?

2-     Introduction part needs to be improved in a way to show the research gap clearly, state the problem simply and show the problems based on facts and figures for Starbucks.

3-     Recent references must be used in the study. Prior literature reviewed in your study looks outdated. You need to strengthen the section with literature/references based on the articles published recently, to convince that your paper examines prior work and has some new/novel paradigm. The authors need to rewrite their introduction, mentioning the rationale behind undertaking this study.

4-     Literature Review-This is bulky with some irrelevant information; it should be summarized under two to three pages. A well-summarized review of related literature would open the glaring gap and give relevance to the intending study. Make the case more strongly in the literature review as to why you're doing this research. It could be much stronger as you can point to truthfulness as an important construct. The paper could use that. You need to strengthen the section with literature/references based on the articles published recently, to convince that your paper

5-     Discussions and implications: I would recommend you highlight how your findings help enrich our understanding. The current version of the discussion section does not sufficiently highlight the contributions of the study

Discussion
I would recommend structuring your discussion in a standard fashion:
1) Brief summary of findings,
2) Theoretical implications,
3) Practical implications,
4) Limitations and avenues for future research

6-     Some of your limitations can be variables like customer relationship quality, social media sites (e.g. Facebook, Instagram, Twitter, Trip Advisor) … Please propose some more research ideas for these variables too. Please see:

-https://hrcak.srce.hr/259223

-https://doi.org/10.1080/13527266.2021.1984279

Reviewer 2 Report

Dear authors,

CSR approach is very important theme in praxis and for firms in Chine 

is important to use social, economical and environmental approach. 

The article is prepared by statistical methods, that are relevant for research.

Reviewer 3 Report

Overall. I found this paper interesting and has potential for publication. However, I would like to suggest the authors make the following amendments.

  1. Abstract: The reference made to Covid-19 period is not suitable to be included in the abstract since it is not been tested. Consider removing it

  2. The author tries to justify the contribution by claiming tthat he evidence from China can reveal the relationship between CSR and FP from different perspectives. The statement is to vague. Further elaboration is needed. 

  3. The contributions of this paper must be explicitly discussed. I dont see the comparison of this paper to others (for example Kao et al (2018)). It should discuss how this paper is different from other papers (at least highlight two papers that are closely related to this paper. It can be from other countries). Highlight the difference between this paper with at least the two other papers.

  4. Table 1 must be rechecked. Use consistent formatting. For example, (1) “ownership concentration” should be “Ownership concentration”. (2) “If the actual controller is the government” change to “A dummy variable that takes value 1 if ..”

  5. The regression models should be written properly. Add the details for the control variables in equations (1),  (2) and (3).

  6. Add relevant literature to CSR, for example

Kamarudin, K.A., Ariff, M.A., & Wan Ismail, W.A. (2022). Product market competition, board diversity and corporate sustainability performance: international evidence. Journal of Financial Reporting and Accounting, 20(2), 233-260.

Kao, E.H., Yeh, C.C., Wang, L.H., and Fung, H.G. (2018). The relationship between CSR and performance: Evidence in China, Pacific-Basin Finance Journal, 51, 155-170,

Muhmad, S.N., Ariff, M.A., Abd. Majid, N., & Kamarudin, K.A. (2021). Corporate governance, product market competition and ESG. Asian Academy of Management Journal of Accounting and Finance, 17(1), 63-91.

  1. Wrong label “ Table 6. Mediation result”. The author does not test mediation.

  2. Finally, the paper should be proofread again.

Reviewer 4 Report

The topic is indeed up to date with the context we are all living in. It has an interesting approach over the CSR reporting and firm performance.

However the original value of this work is difficult to be found. The authors should clearly highlight which are the original value of this research.

Abstract: The abstract is well written. It includes synthetically the brief introduction into the researched topic, sample, methods, findings and short conclusions.   

Introduction: In the introduction, you should clearly underline the gap found regarding the researched topic and also include the raised question the developed research.

Literature review and hypotheses development:

The literature review is well organized, very well documented and follows a logical path reaching the hypothesis of the paper.

Research Design: Relate in a few words, the models used to process the statistics.

Results: Should mention the source of the tables/ figures.

Conclusions: The conclusions are well written, resuming the results attained in the conducted research.

References appear to be outdated. The authors should update they to 2023.

In order to substantiate the research, you can have a look over the following articles:

1)      https://doi.org/10.3390/su12187545

2)     https://doi.org/10.3846/16111699.2013.834841

3)     https://doi.org/10.1186/s43093-021-00075-8

4)     https://doi.org/10.1080/1331677X.2021.2017318

5)     https://doi.org/10.1080/1331677X.2022.2080745

Reviewer 5 Report

Synopsis:

The study examines the impact of Corporate Social Responsibility (CSR) on Financial Performance (FP) and the moderating effect of media attention on this relationship. The study results indicate that (1) companies with good CSR performance show a high level of FP; (2) The higher the media's attention to the company, the better the performance of CSR.

Abstract is not clear.  Please, write a more focused, purposeful abstract that covers the key points such as study objectives, methods used, results, and implications. The current abstract seems long and not purposeful.

Comments and suggestions:

-    The authors need to articulate their research questions/objectives, identify the potential theoretical, background and theoretical motivation or gaps, and explain how your study contributes to the literature since there are ample of studies that examined the association between CSR and FP. What is the new thing we should know about the association between CSR and FP? Authors should straightway start with highlighting these important issues and clearly state the contribution of the study. The authors need to elaborate more on the effect of the moderating variable (i.e., media attention).   

-    In the introduction, there are many sentences without citations for example:

“Chinese companies are contributing more and more to the international market. Although China has turned to a stage of high-quality development, a series of problems, such as medical health, food safety, and old-age care, which are close to the people's livelihood, have become increasingly prominent, and enterprises have a weak awareness of fulfilling and disclosing CSR”. Who said these, please check thorough the article.  

-    The authors claim that one of their main contributions is choosing China as a research context. I think this is not a contribution as there ae many studies that examine the association between CSR and FP. The authors need to elaborate more on the study contribution.

-    We all know about the definition of CSR; the authors need to review and discuss the theoretical background of the study.

-    The authors need to justify why they choose the time period from 2014 to 2020 to collect the study samples.

-    No justification for the type of regression used in the study and whether the regression assumptions are tested or not. Why the authors didn’t use the panel data method since they have almost 7 years.  

-    The authors mentioned in the abstract that they controlled for endogeneity, but in the methodology and discussion nothing related to the endogeneity tests or control. 

-    The conclusion part needs extensive work and reorganization to highlight the importance of study, objective, findings, and implications.

-    The implications are too general and did not offer anything new or interesting. I would like to see the differences and similarities between the findings of this research and those of previous research. How do the findings of this research differ from previous research?

-    The paper should be proofread by an expert to remove grammatical mistakes and inconsistencies.

Reviewer 6 Report

Investigating the relationship between CSR and financial issues is popular; this manuscript can make a valuable contribution to the knowledge base by the highlight of the media attention. The literature review and the sources involved are acceptable, and the structure of this part follows the standards.

There are some issues to be developed before publishing:

1. Hypothesis development is acceptable, and statistical analysis is in line with the hypotheses, but a clear response to the questions is missing.

2. The statistical analysis has a great emphasis, and many details are presented, but the discussion is short. I suggest adding more details and changing the order. First, show an understanding of your results and ideas, and compare this to former results only after it.

3. Proofreading of the text is required. Several typos and editing issues have remained in the text. Some of these may come from using reference management software (missing and extra spaces).

I can recommend the publication after minor amendments following the instructions above.

Reviewer 7 Report

The paper is an overall acceptable paper, the consideration of media coverage is good as it gains more attention from scholars in the field.

Good paper, however, some points need a little bit more clarification.

1-  Statement of hypotheses 1 and 2 need to be better articulated.

2- Quantification of the Moderator variable needs to be better explained (more details).

Round 2

Reviewer 1 Report

1.      Recent references must be used in the study. Prior literature reviewed in your study looks outdated. You need to strengthen the section with literature/references based on the articles published recently, to convince that your paper examines prior work and has some new/novel paradigm. The authors need to rewrite their introduction, mentioning the rationale behind undertaking this study.

2.      Without a strong engagement in recent studies, the study seems generic and cursory, the arguments seem patched and do not extend much beyond existing literature, and the paper does not seem to provide the type of deep, novel theoretical contributions we typically seek at novel research. The innovation of the paper is not enough; the research question is not clear. Make strong discussions about why the research is important, what motivations are required to complete the research, and the gaps you want to search?

3.      I could not understand the gaps or contributions of this research. The gap and contribution part make the case more strongly depending on the recent literature review about the topic and the needed gaps. they should make a better effort in contextualizing the research gap with more (recent) literature addressing the main authors' topic.  What is the new thing we should know about the relationship between CSR and Financial Performance?

4.      Highly simplistic and obvious your limitations, managerial implications and further research.

5.      Discussions and implications: I would recommend you highlight how your findings help enrich our understanding. The current version of the discussion section does not sufficiently highlight the contributions of the study.

Discussion
I would recommend structuring your discussion in a standard fashion:
1) Brief summary of findings,
2) Theoretical implications,
3) Practical implications,
4) Limitations and avenues for future research

Reviewer 5 Report

The authors didn't address the following comments:

- The authors need to justify why they choose the time period from 2014 to 2020 to collect the study samples.

- No justification for the type of regression used in the study and whether the regression assumptions are tested or not. Why the authors didn’t use the panel data method since they have almost 7 years.  

- The authors mentioned in the abstract that they controlled for endogeneity, but in the methodology and discussion nothing related to the endogeneity tests or control.  

Round 3

Reviewer 1 Report

Thank you